# Subclinical Hypothyroidism and Gestational Hypertensive Disorders in a Cohort of Romanian Pregnant Women with Gestational Diabetes Mellitus: A Pilot Study

**DOI:** 10.3390/biomedicines12112587

**Published:** 2024-11-12

**Authors:** Muntean Mihai, Săsăran Vladut, Pop Gheorghe Lucian, Muntean Elena Irina, Nyulas Victoria, Mărginean Claudiu

**Affiliations:** 1Department of Obstetrics and Gynecology 2, University of Medicine Pharmacy Science and Technology George Emil Palade of Târgu Mureș, 540142 Târgu Mureș, Romania; munteanmihai@yahoo.com (M.M.); marginean.claudiu@gmail.com (M.C.); 2Department of Obstetrics and Gynecology, Polizu Hospital Clinic, University of Medicine Pharmacy Carl Davila, 050474 Bucharest, Romania; popluciangh@icloud.com; 3Algcocalm SRL, 540360 Târgu Mureș, Romania; irinagauca@yahoo.com; 4Department of Informatics and Medical Biostatistics, University of Medicine Pharmacy Science and Technology George Emil Palade of Târgu Mureș, 540142 Târgu Mureș, Romania; victoria.rus@umfst.ro

**Keywords:** gestational diabetes mellitus, subclinical hypothyroidism, gestational hypertension, perinatal outcome

## Abstract

Objective: This study aimed to explore whether subclinical hypothyroidism (SCH) treated with levothyroxine in pregnancy complicated by gestational diabetes mellitus (GDM) is associated with an increased risk of gestational hypertensive disorders (GHDs) (gestational hypertension and preeclampsia). Methods: 96 pregnant women with GDM were enrolled in this study and grouped as per the European Thyroid Association criteria into the SCH (n = 21) and euthyroid groups (n = 75). All subjects were tested for anthropometric parameters, maternal glucose homeostasis parameters, lipid levels, thyroid function tests, and blood pressure. All GDM pregnant women received nutritional and insulin therapy where needed, and the SCH group received levothyroxine treatment. Then, the maternal and newborn outcomes were compared. Data were analyzed using Student’s *t*-test, Mann–Whitney U, and Chi-square tests wherever applicable. *p* values of <0.05 were considered significant. Results: Patients with GDM and SCH had a pre-pregnancy BMI and BMI at inclusion in the study smaller than those of the euthyroid group (*p* = 0.0004, *p* = 0.0009). There were no significant differences between groups regarding the incidence of GHD, preterm prelabor rupture of membranes (PPROMs), macrosomia, low birth weight, and fetal distress (*p* > 0.05). Patients with GDM and SCH treated with levothyroxine had more premature delivery than the euthyroid group (*p* = 0.03). Conclusions: Subclinical hypothyroidism treated with levothyroxine in women with GDM does not increase the risk of gestational hypertensive disorders, but is associated with increased risk for prematurity.

## 1. Introduction

Pregnancy is a challenging test for maternal thyroid function. Placental deiodinase type 3 degradation of thyroid hormones, renal iodide clearance, and serum thyroxine-binding globulin levels increase during pregnancy [1]. The consequences are an increased need for thyroid hormones T3 and T4 production throughout pregnancy, increased serum concentrations of total T4 and T3, increased free T4 (FT4) levels, decreased serum thyroid stimulating hormone (TSH) and thyroglobulin levels, and in iodine-deficient areas, thyroid hormone production decrease [2].

Thyroid hormones T4 and T3 are essential for fetal growth and neurodevelopment during pregnancy. The fetal thyroid gland begins to produce adequate quantities of thyroid hormones during the second trimester. Thus, the fetus depends on maternal thyroid hormones during pregnancy, especially during the first trimester [3].

Hypothyroidism’s causes could be iodine deficiency, Hashimoto’s thyroiditis, post thyroidectomy, secondary to medication (amiodarone), suppurative thyroiditis, lymphocytic hypophysitis, Sheehan’s syndrome, and congenital hypothyroidism [4].

Subclinical hypothyroidism (SCH) represents an early stage of thyroid failure, typically diagnosed in the general population by a TSH level below 10 mU/l and a normal FT4 level [5,6]. The diagnosis is biochemical because the symptoms are mild and non-specific [7]. In recent years, significant progress has been made in the methods of diagnosing and treating thyroid disorders that occur during pregnancy [8].

A serum TSH level higher than the upper limit of the pregnancy-trimester-related reference range associated with an average serum FT4 concentration is defined as SCH in pregnancy [9]. The incidence of SCH in pregnancy in Europe is 10% [10]. The European Thyroid Association recommends using TSH levels higher than 2.5 mU/l for the first trimester, 3.0 mU/l for the second trimester, and 3.5 mU/l for the third trimester of pregnancy for diagnosing SCH when trimester-specific reference ranges are not available [9]. The Romanian Society of Endocrinology recommends using European Thyroid Association and American Thyroid Association recommendations for diagnosing and treating SCH in pregnancy [10]. In different countries, the incidence of this condition may vary depending on which cut-off values are used to define SCH.

Women with SCH with or without positive thyroid peroxidase antibodies (TPOAb) are at high risk of abortion [11], premature rupture of membranes, placental abruption, newborn death [12], gestational hypertension [13], preeclampsia [11,14,15], and gestational diabetes mellitus (GDM) [11,16], but some other studies have not found that SCH with or without positive TPOAb is associated with pregnancy complications [17,18,19].

Additionally, the impact of levothyroxine treatment in pregnant women with SCH is not clear. Some studies show that levothyroxine treatment may lower the risks of abortion and stillbirth [20], gestational hypertension, preterm delivery [21,22] GDM [23], low birth weight [24], and neonatal death [22]. Others have found that levothyroxine therapy does not influence obstetrical, neonatal outcomes, and childhood IQ score or neurodevelopmental outcomes [25,26], and others have found that treatment with levothyroxine is associated with a higher rate of preterm labor in women with SCH [20].

The incidence of GHD in Romania is 3.7% [27]. GHD is associated with a higher incidence of perinatal complications, like preterm birth, SGA neonates, stillbirths, and delivery by cesarean section [27].

Another complication of pregnancy is gestational diabetes mellitus (GDM) [28], which has a prevalence of 10.9% in Europe [29]. The American Diabetes Association (ADA) defined GDM as any glucose intolerance diagnosed during the second or third trimester of pregnancy [30]. GDM occurrence is associated with some risk factors, like heredo-colateral history of type 2 diabetes mellitus (T2DM), maternal age above 30–35 years, increased body mass index (BMI) greater than 25 kg/m^2^, personal history of macrosomia (newborn with a birth weight > 4000 g), hypothyroidism, smoking, and other ethnicities than Caucasian [31,32]. Its pathophysiology involves insulin resistance, β-cell dysfunction, chronic inflammation, and low circulating adiponectin levels [32,33,34]. GDM induces a wide range of complications for mothers and newborns. Some of the short-term complications are polyhydramnios, gestational hypertension [35], pre-eclampsia, operative delivery, shoulder dystocia, birth canal trauma, macrosomia, neonatal hypoglycemia, and jaundice [28]. Long-term complications include the mother’s increasing risk of T2DM, metabolic syndrome, and cardiovascular disease, and the offspring’s long-term complications are increased risks of hyperglycemia, T2DM, obesity, and cardiovascular disease [28].

The prevalence of SCH and GDM described in the literature is 4.18% [19].

Although there are data from Romania on iodine deficiency and its treatment in pregnant women [36], there are no data on subclinical hypothyroidism or the evolution of these patients under treatment with levothyroxine.

This study aimed to explore the impact of SCH treated with levothyroxine on gestational hypertensive disorders (GHDs) (gestational hypertension and preeclampsia) in pregnant women diagnosed with GDM.

## 2. Material and Method

### 2.1. Study Design

The University of Medicine, Pharmacy, Science and Technology “G. E. Palade” of Târgu–Mures Ethics Committee authorized this study (decision number 1557/2022) in accordance with the principles of the Declaration of Helsinki (1964).

### 2.2. Description of Study Area and Duration of Study

This prospective case–control study was conducted at Obstetrics and Gynecology Clinic 2 in Târgu Mureș, Romania, between 1 January 2022 and 15 October 2024. Our cohort was consecutively selected and included pregnant women diagnosed with GDM.

### 2.3. Inclusion and Exclusion Criteria

The inclusion criteria were as follows: singleton pregnancy between 24 and 28 weeks of gestation (WG), Romanian ethnicity, diagnosis of GDM, age between 18 and 46 years, compliance with follow-up conditions, and delivery at the Obstetrics and Gynecology Clinic 2 Târgu Mureș. The exclusion criteria included the following: known patients with hypothyroidism with or without positive TPOAb, diagnosed before pregnancy or in the first trimester of pregnancy; patients with type 1 diabetes or T2DM diagnosed before pregnancy; pregnancies with chromosomal or fetal malformations; cases of intrauterine fetal death; and those who lacked informed consent.

Before enrollment in this study, written informed consent was obtained from all patients.

After applying the inclusion and exclusion criteria, we included 96 pregnant women with GDM in this study. Based on their TSH and FT4 results at the time of the oral glucose tolerance test (OGTT), they were divided into two groups: 75 euthyroid pregnant women with GDM and TSH levels < 3.0 mU/l and 21 pregnant women with GDM and TSH levels ≥ 3.0 mU/l, based on European Thyroid Association recommendations (Figure 1).

### 2.4. Diagnosis of GDM

The International Association of Diabetes and Pregnancy Study Groups criteria were used to diagnose GDM. One or more abnormal glucose values above ≥92 mg/dl (≥5.2 mmol/l), 1 h ≥ 180 mg/dl (>10 mmol/l), or 2 h ≥ 153 mg/dl (>8.5 mmol/l) [37] were used for GDM diagnosis. Patients diagnosed with GDM were advised to engage in moderate physical activity, follow nutritional therapy, and monitor their glycemic status through three daily assessments of fasting and postprandial glucose levels for two weeks. A diabetologist prescribed insulin therapy at a dosage of 0.7–1.0 units/kg of body weight per day for women who did not achieve glycemic control through diet. The target glucose levels were fasting below 95 mg/dl and postprandial levels below 120 mg/dl at 2 h after eating [38]. The pregnant women continued to monitor their glucose levels until birth under the surveillance of the diabetologist.

### 2.5. Diagnosis of SCH

According to the European Thyroid Association, SCH is diagnosed if TSH levels in the second trimester (24–28 WG) are 3.0 mU/l [9]. Levothyroxine treatment was initiated after SCH was diagnosed at 24–28 WG at the OGTT time. An endocrinologist treated patients with SCH by administering 1.2 μg/kg/day of levothyroxine to achieve a TSH level below 2.5 mU/l. All patients undergoing levothyroxine treatment had their TSH levels checked at 4-week intervals. Their levothyroxine dosage was adjusted according to the endocrinologist’s advice to achieve a TSH level of <3 mU/l in the second trimester and <3.5 mU/l in the third trimester of pregnancy. As our study did not assess thyroid peroxidase antibodies (TPOAb), we are considering including TPOAb testing in future studies.

After inclusion in the study, all pregnant women were scheduled for appointments for routine prenatal surveillance every two weeks or sooner if clinically indicated until delivery as part of routine prenatal care.

Upon inclusion in this study, at 24–28 WG, we collected demographic characteristics like maternal age, gestation, parity, smoking, chronic hypertension, and heredo-collateral history of T2DM; we performed anthropometric measurements and obtained a fasting blood sample. At delivery, we reviewed the medical records of each patient. We recorded information on pregnancy complications (gestational hypertension, preeclampsia, placental abruption, premature delivery < 37 weeks, preterm prelabor rupture of membranes (PPROM) < 37 weeks, macrosomia, low birth weight, and fetal distress), mode of delivery, and anthropometric measurements of the mothers and newborns.

### 2.6. Maternal and Neonatal Complications

We defined maternal and neonatal complications as follows:

Gestational hypertension (GH) = a systolic blood pressure reading of 140 mm Hg or higher, or a diastolic blood pressure reading of 90 mm Hg or higher, or both, on two separate occasions at least 4 h apart after 20 weeks of pregnancy in a woman who previously had normal blood pressure [39].

Preeclampsia = gestational hypertension and proteinuria ≥ 300 mg/24 h urine collection [39].

Placental abruption = the premature separation of the placenta from the uterine wall after 20 WG and before birth [40].

Premature delivery = birth at less than 37 completed WG, according to ICD 10 definitions (O60).

Preterm prelabor rupture of membranes (PPROM) = membrane rupture before labor that occurs before 37 WG [41].

Macrosomia = birth weight ≥ 4000 g [42].

Low birth weight = birth weight below 2500 g [43].

Fetal distress = Cardiotocographic signs of fetal compromise in labor [44].

### 2.7. Anthropometric Measurements

In this study, we included maternal pre-pregnancy BMI, BMI at 24–28 WG, and at birth. Maternal pre-pregnancy BMI (kg/m^2^) was calculated from pre-pregnancy medical records or where it was impossible to patient-reported pre-pregnancy weight and height measured during the first prenatal visit during the first trimester of pregnancy. BMI at 24–28 WG was measured during OGTT, and weight at birth was measured at admission to the hospital before birth. The patient’s height (cm) was measured using a wall tape measure without shoes and estimated to the nearest 1 mm. The patient’s weight was measured using a Beurer PS digital scale (Beurer Gmbh, Ulm, Germany) and subtracting 0.5 kg as the weight of the patient’s clothing. Newborn anthropometric measurements (weight, length, and ponderal index at birth) were performed in the first 30 min after birth. Ponderal index was calculated as 100 × [birthweight (g)/length (cm^3^)].

Systolic blood pressure (SBP) and diastolic blood pressure (DBP) were measured according to the American College of Obstetricians and Gynecologists Practice Bulletin no 202 recommendations at the inclusion in this study [39].

### 2.8. Biochemical Analyses

We obtained maternal blood samples during OGTT in a fasting state between 8 and 10 a.m. using the classic peripheral blood collection technique before any treatment for GDM was administered.

We measured the fasting glucose level, 1 h glucose level, and 2 h glucose level after administering 75 g of glucose orally from maternal blood at 24–28 WG. Additionally, we also assessed TSH, FT4, C-reactive protein (CRP) level, HbA1c, HOMA-IR, total cholesterol (TC), high-density lipoprotein cholesterol (HDL-C), low-density lipoprotein cholesterol (LDL-C), and triglycerides (TG).

The Atellica Solution CH 930 device (Siemens Healthcare GmbH, Forchheim, Germany) was used to determine glucose levels in blood using spectrophotometry, CRP and glycosylated hemoglobin using turbidimetry, TC, HDL-C, LDL-C, and TG values using photometry, and TSH and FT4 using chemiluminescent microparticle immunoassay. The formula used to estimate IR HOMA was: [(fasting insulin (mU/l) × fasting glucose (mmol/l)]/22.5 [45].

### 2.9. Statistical Analysis

GraphPad Prism version 9.0 software (GraphPad Software, Boston, MA, USA) was used for statistical analyses. The median (IQR) and mean ± standard deviation were used to express continuous data. Percentages were used to express categorical data. When dealing with regularly distributed continuous variables, Student’s *t*-test was utilized, and when dealing with non-normally distributed data, the Mann–Whitney test was chosen. Clinical characteristics of SCH and control participants were compared using the chi-square test or Fisher’s exact testing for categorical data. A two-tailed *p*-value of less than 0.05 was accepted as statistically significant.

The power sample was calculated with G*Power sample size calculation software (version 3.1.9.7.), which indicated that for a power of 80% and a high effect size of 0.7, a sample ratio of 1/3 requires 72 patients for the control group and 21 patients for the testing group.

## 3. Results

### 3.1. Demographic, Anthropometric Characteristics, and Laboratory Results at 24–28 Weeks of Pregnancy

Our study found that patients with SCH were older than patients from the euthyroid group, but without significant differences (*p* = 0.25). There were no significant differences between groups regarding gestation (*p* = 0.61), parity (*p* = 0.55), smoking (*p* = 0.14), heredo-colateral history of T2DM (*p* = 0.79), chronic hypertension (*p* > 0.99), and gestational age at inclusion in this study (*p* = 0.73). Patient characteristics are shown in Table 1.

We found that patients from the euthyroid group had higher pre-pregnancy BMI (28.54 kg/m^2^, 95% CI 27.25–29.84 kg/m^2^ (*p* = 0.0004)), BMI (31.14 kg/m^2^, 95% CI 29.9–32.39 kg/m^2^ (*p* = 0.0009)), and CRP values (0.8 mg/dl, 95% CI 0.65–0.94 (*p* = 0.01)) at inclusion in this study compared with patients from the SCH group. There were no differences between groups regarding SBP, DBP values, glucose homeostasis parameters, and lipid values. The serum values of TSH in the SCH group were significantly greater than those in the euthyroid group (3.82 mU/l, 95%CI 3.5–4.13 (*p* < 0.0001)). There were no differences in FT4 values between the groups (*p* = 0.27).

### 3.2. Maternal and Newborn Demographic, Anthropometric Characteristics, Results at Birth

There was no difference between groups concerning gestational age at birth (*p* = 0.42), cesarean section rate (*p* = 0.45), newborn weight at birth (*p* = 0.58), and newborn ponderal index (*p* = 0.43). However, patients from the euthyroid group had a higher BMI at birth than patients from the SCH group (33.08 kg/m^2^, 95% CI 31.8–34.36 kg/m^2^ (*p* = 0.005)). The patient’s characteristics are shown in Table 2.

### 3.3. Maternal and Perinatal Outcomes

The incidence of GH in the euthyroid group was 17.33%, and in the SCH group, it was 28.57%.

We found no differences between the groups regarding the incidence of GHD, PPROM, macrosomia, low birth weight, and fetal distress. In both groups, there were no cases of placental abruption. However, patients from the SCH group had more premature deliveries (*p* = 0.03) than those in the euthyroid group. The patients’ characteristics are shown in Table 3.

Regarding the relationship between TSH levels at inclusion in the study and premature birth, we found that TSH levels above 4.2 mU/l had an excellent predictive ability to predict premature delivery with an AUROC of 0.94 (95% CI 0.75 to 0.99) (Figure 2).

Table 4 shows the results of multivariate regression analyses assessing premature delivery. We found that a higher TSH level at study inclusion had an OR of 7.46 for premature delivery. Increased maternal age and GH were associated with OR of 1.46 and 7.0 for premature delivery, but without statistical significance (*p* = 0.06 and respectively 0.14). Figure 3 shows the forest plot for premature delivery.

## 4. Discussions

Pregnancy can affect thyroid function, and there are changes in TSH, FT4, and FT3 serum levels during pregnancy as pregnancy advances [2]. Thyroid hormones are essential for fetal growth and neurodevelopment [3]. There are variations in thyroid hormone levels according to ethnicity, country, and detection methods [9]. Iodine deficiency is the most important cause of hypothyroidism in Eastern Europe [46]. There is no trimester-specific reference range for thyroid hormones for Romanian pregnant women, so we used a trimester-specific reference range from the European Thyroid Association Guidelines to diagnose SCH in the second trimester of pregnancy [9].

Pregnant women with SCH can develop GDM [11,16] and hypertensive disorders [11,14,15], and also GDM pregnant women are at increased risk of hypertensive disorders [28], increasing the risk for adverse outcomes for the pregnant women, fetus, and newborn. Additionally, there is conflicting evidence regarding the usefulness of levothyroxine treatment of SCH in pregnancy [20,21,22,23,24,25,26].

This is the first study to evaluate the impact of SCH and levothyroxine treatment on maternal and neonatal outcomes of a cohort of Romanian GDM-diagnosed pregnant women.

In our cohort, there were no differences between groups regarding age at inclusion in this study (32.71 ± 3.88 vs. 31.49 ± 5.33 years, *p* = 0.25). Pregnant women with GDM and SCH from the Durcame et al. [47] study were 28.8 ± 4.6 years. This difference can be explained by postponing getting pregnant in our study group and increasing thyroid dysfunction caused by aging [48]. We did not find any differences between the groups regarding gestation, parity, smoking, heredo-colateral history of T2DM, and gestational age at the inclusion in this study. The same results were found by Durcame et al. [47] and Chen et al. [49].

Pregnant women with GDM and SCH from our cohort were overweight before pregnancy and at inclusion in this study. The pregnant women from the euthyroid group had significantly greater pre-pregnancy BMI (*p* = 0.0004) and BMI at inclusion in this study (*p* = 0.0009). Additionally, pregnant women from the SCH group had substantially lower CRP values than pregnant women from the euthyroid group (*p* = 0.01). The greater CRP serum values in the euthyroid group than those in the SCH group are in the context of low-grade chronic inflammation of greater BMI values of these patients [50]. On the contrary, Durcame et al. [47] found in their study that pregnant women with SCH were obese grade 1 at admission to the study, and the pregnant women from the euthyroid group were overweight (*p* = 0.20). Additionally, Croce et al. [51] found that TSH was significantly higher in obese women with negative TPOAb compared with overweight and normal-weight pregnant women in the first trimester of pregnancy. However, Knight et al. [52] did not find a positive correlation between SCH and BMI in the first trimester of pregnancy.

We did not find any differences between groups regarding SBP, DBP, glucose metabolism parameters, or lipid values in the second trimester of pregnancy. Only triglyceride values in both groups exceeded the reference values for healthy pregnant women in the second trimester [53] Ryckman et al. [54] showed in their meta-analysis that triglyceride levels were elevated from the first trimester until the third trimester of pregnancy of women who will develop GDM during pregnancy, suggesting the role of metabolic dysfunction early in pregnancy or even before pregnancy in GDM appearance. Wang et al. [55] found in their study that pregnant women with SCH and dyslipidemia in the first trimester of pregnancy were older (>35 years) or overweight/obese and had a greater incidence of GH, preeclampsia/eclampsia, GDM, and low birth weight. Lower thyroid hormone and higher TSH values encountered in hypothyroidism increase T-cholesterol, LDL-cholesterol, and triglyceride values by decreasing lipoprotein lipase function, reducing the activity of cholesterol 7α-hydroxylase and ATP-binding cassette transporter G5/8, and increasing Niemann–Pick C1-like 1 protein concentration; however, these changes are seen in patients with overt hypothyroidism [56].

Regarding TSH levels in the second trimester of pregnancy, they were significantly higher in the SCH group (*p* < 0.0001), with no differences for FT4 levels (*p* = 0.27). Regarding the TSH levels used to diagnose SCH, we used the European Thyroid Association’s reference range (TSH > 3.0 mU/l in the second trimester of pregnancy). In the literature, there is no general agreement regarding the TSH levels used to diagnose SCH; more thresholds are used; therefore, there are contradictory conclusions. Cakmak et al. [11] used values of >2.5 mU/l and Durcame et al. [47] used the reference levels of the European Thyroid Association, but Fatima et al. [57] used the range of 4.4–10 mU/l to diagnose SCH.

Our patient groups had no differences concerning gestational age, newborn weight, or ponderal index at birth. Regarding mode of delivery, 66.66% of patients from the SCH group gave birth by cesarean section, but without statistical difference from the euthyroid group (*p* = 0.45). Also, Durcame et al. [47] found no differences between GDM pregnant women with and without SCH regarding gestational age at birth, mode of delivery, and newborn weight.

Regarding maternal complications during pregnancy, we did not find significant differences between groups regarding GH, preeclampsia, PPROM, and placental abruption.

The rate of GH in our cohort of GDM pregnant women with and without SCH (28.57% and 17.33%) was more significant than the incidence of GH found by Walker et al. [58] in a cohort of pregnant women without GDM or SCH (6.3%). This difference is because our cohort patients were older and had a greater BMI than those in the Walker et al. [58] group. In our cohort, we did not have cases of placental abruption.

Regarding GDM, GH, and preeclampsia, Kvetny et al. [59] and Bryson et al. [60] found that GH and preeclampsia appeared more frequently in women with GDM, and BMI was higher in women with GH or preeclampsia and GDM compared with controls [60]. Bryson et al. [60] found that GDM increased the risk of developing hypertensive disease during pregnancy by 1.5-fold, and this was greater in different ethnic groups with less prenatal care. From pathophysiologic points, GDM and hypertensive disorders of pregnancy share similar abnormalities, like insulin resistance, central obesity, dyslipidemia, oxidative stress, endothelial dysfunction, chronic inflammation (neutrophil activation, abnormal cytokines), and abnormal levels of adipokines (low levels of adiponectin, high levels of leptin) [61].

Regarding the role of SCH in hypertensive disorders of pregnancy (HDP), some researchers, like Cakmak et al. [11], found a significantly higher incidence (*p* = 0.004) of GH in a cohort of 930 pregnant women with untreated SCH compared with 7986 control patients, matched for age and BMI; Han et al. [14], in their meta-analysis, found that a TSH level > of 3 mU/l, regardless of trimester of pregnancy, was associated with increased risk by 1.67 times of HDP. Lai et al. [13] suggested that SCH in early pregnancy is a risk factor for GH later in pregnancy. Tolosa et al. [15], in their meta-analysis, found that SCH, defined as TSH levels above the 97.5th percentile of the reference range and FT4 levels within the reference range (2.5th–97.5th percentile), was associated with a higher risk of preeclampsia. Some previous studies concluded that alteration in thyroid hormones seen in SCH could increase the risk of hypertensive disorders of pregnancy by endothelial dysfunction resulting from a reduction in nitric oxide availability [62], impaired cardiac contractility, impaired diastolic function, and increased serum cholesterol [63]. On the contrary, other studies did not show that SCH was associated with an increased risk of gestational hypertension. Furukawa et al. [17] did not find an increased risk of GH in 167 pregnant women with SCH (median TSH 3.4 mU/l) and a mean BMI of 22 ± 3.7 Kg/m^2^. The fact that they did not have obese patients in the SCH group is the reason why they did not find an increased risk of GH associated with SCH in pregnancy.

Regarding premature delivery, we observed significantly more premature delivery in the SCH group (*p* = 0.03). Stagnaro-Green et al. [64] found an increased risk of very preterm birth (<32 weeks) associated with high TSH levels. Still, they did not observe the same association in the case of moderately preterm deliveries. Lee et al. [65] found that a TSH level > 4 mU/l was associated with a 2.17-fold increased risk of prematurity. Also, we found that a level of TSH > 4.2 mU/l in the second trimester of pregnancy was associated with an OR of 7.46 (95% CI 1.04–53.36) for premature birth. One explanation for higher rates of premature births in our SCH group could be related to vascular disease etiology: endothelial dysfunction in the context of lower nitric oxide levels in SCH patients [62] and the failure of the physiologic transformation of the myometrial segment of spiral arteries, along with placental vascular lesions induced by maternal vascular poor perfusion [66,67]. Another explanation for a higher premature birth rate could be the association of GDM and SCH with gestational hypertensive disease, along with an increased rate of medically induced premature delivery [68].

Regarding newborn outcomes, we found no differences between groups in the incidences of macrosomia, low birth weight, and fetal distress. Durcame et al. [47] also found no differences between mothers with and without SCH in macrosomia and low birth weight incidence.

All GDM pregnant women in our study received nutritional and insulin therapy where needed. Pregnant women diagnosed with SCH were treated by an endocrinologist with 1.2 μg/kg/day of levothyroxine until birth to achieve a serum TSH level below 2.5 mU/l. We did not find an increased risk of GHD in our SCH group treated with levothyroxine, but our results must be interpreted with caution due to our small sample of GDM patients with SCH. We found an increased risk of premature birth in the SCH group treated with levothyroxine. Sankoda et al. [69] found in their meta-analysis that levothyroxine treatment during pregnancy with SCH was associated with a reduced risk of preterm birth when TSH levels were >4.0 mU/l, but no significant effect when TSH levels were between 2.5 and 4.0 mU/l. According to our findings, Sitoris et al. [23] found that pregnant women with untreated SCH (TSH > 3.74 mIU/l) and without TPOAb had a higher rate of GDM and pre-eclampsia compared with women without SCH. Additionally, they found that if the levothyroxine treatment was initiated even after the first trimester, the pregnant women did not develop these complications. In their meta-analysis, Ding et al. [21] found that pregnant women with SCH diagnosed after American Thyroid Association criteria (TSH > 4.0mIU/l) and treated with levothyroxine had lower risks of pregnancy loss, preterm birth, and gestational hypertension. In contrast, Yamamoto et al. [25] in their meta-analysis, which included only three randomized trials, found no benefit of levothyroxine treatment on gestational hypertension, pre-eclampsia, preterm delivery, mode of delivery, neonatal intensive care unit admission, birth weight, and gestational age at delivery.

We did not assess iodine levels and TPOAb status during the second trimester of pregnancy, which can impact thyroid function and pregnancy outcome as a confounding factor. Shenhav et al. [70] found that iodine deficiency in the third trimester of pregnancy can alter thyroid function, and He et al. [71] found that TPOAb positivity in pregnant women with and without thyroid dysfunction was associated with an increased risk of preterm delivery.

The principal strength of our study is that the same multidisciplinary team managed all women with GDM and SCH in our center throughout the study. This avoided significant variation regarding diagnosing GDM and SCH, the need for treatment (insulin and levothyroxine therapy), and perinatal management. We utilized standardized protocols to minimize outcome variations due to differences in clinical practice. We measured weight and height during the second trimester and at birth for BMI calculation instead of relying on the weight and height reported by the patient. This study is one of the first conducted in Romania to explore the GHD in pregnant women diagnosed with GDM and SCH treated with levothyroxine and underline that GDM patients with SCH treated with levothyroxine have no increased risk of GHD, PPROM, macrosomia, and LBW, but have a higher risk of premature delivery compared with GDM patients without SCH.

Our study has some limitations. All our patients were from the same area, and the small sample may have reduced the statistical power of our research. We did not assess the iodine and TPOAb status of our patients as iodine deficiency in pregnancy can influence thyroid function, and positive TPOAb during pregnancy can increase the risk of adverse pregnancy outcomes, including premature birth. We did not have long-term information about the neurodevelopment of the offspring. This study reflects our experience with the interdisciplinary approach of women with GDM and SCH. Our findings can only be applied to other perinatal centers that employ similar practices. The rate of GH in our GDM with SCH group (28.57%) is higher than that in the literature (10.9%) [72] and should be explained by the small sample of women with SCH (*n* = 21) in our study.

Future research could explore the TPOAb status and thyroid function of GDM patients since autoimmune disorders increase the risk of GDM. Another potential direction is to conduct long-term follow-up studies on the neurodevelopment of children born to mothers with GDM and SCH.

## 5. Conclusions

In conclusion, our results suggest that subclinical hypothyroidism treated with levothyroxine in women with GDM does not increase the risk of gestational hypertensive disorders but is associated with an increased risk of prematurity. Given the small sample size of this study’s patients, our findings need to be confirmed in large prospective studies.

## Figures and Tables

**Figure 1 biomedicines-12-02587-f001:**
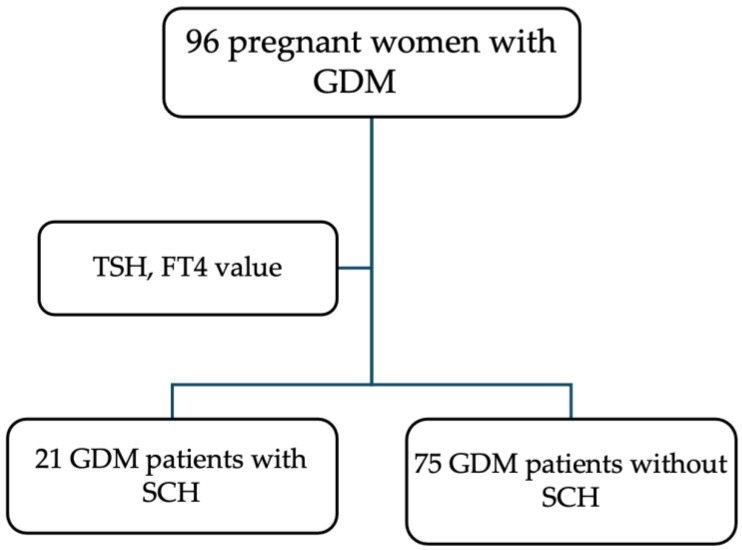
Flow chart of study participants at 24–28 WG. Note: GDM = gestational diabetes mellitus, FT4 = free T4, SCH = subclinical hypothyroidism, TSH = thyroid stimulating hormone.

**Figure 2 biomedicines-12-02587-f002:**
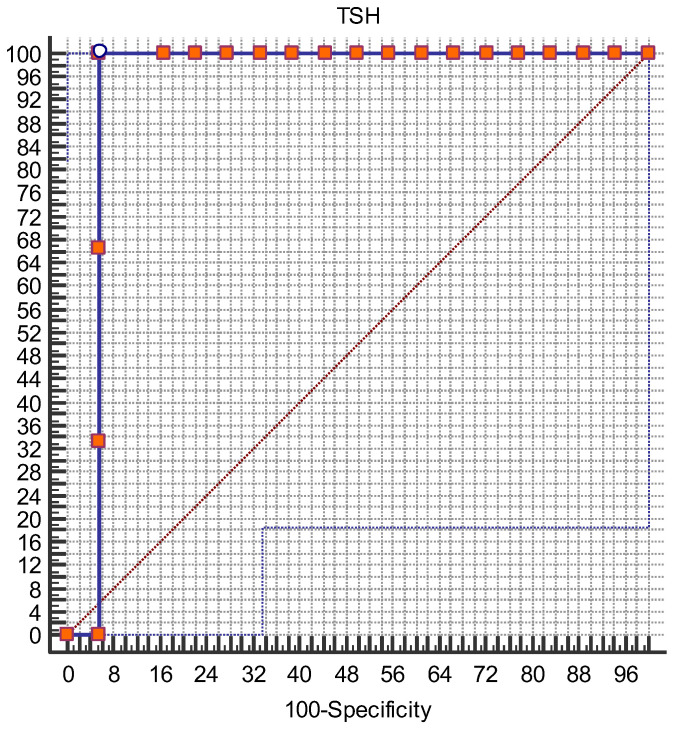
ROC curve for premature birth. Note: ROC curve for TSH levels at 24–28 WG for prediction of premature birth.

**Figure 3 biomedicines-12-02587-f003:**
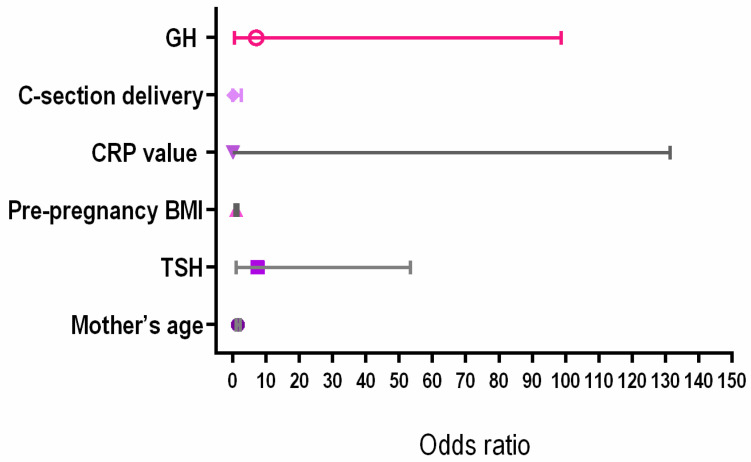
Forest plot for premature delivery. Note: BMI = body mass index, CRP = C reactive protein, C = cesarean, TSH = thyroid stimulating hormone.

**Table 1 biomedicines-12-02587-t001:** Demographic and anthropometric characteristics, and laboratory results at 24–28 weeks of pregnancy.

Parameters	Euthyroid Group n = 75	SCH Group n = 21	*p*-Value
Age, years, Mean, (SD)	31.49 ± 5.33	32.71 ± 3.88	0.25
Gestation, Mean, (SD)	2.47 ± 1.15	2.46 ± 1.59	0.61
Parity, Mean, (SD)	1.87 ± 0.82	1.66 ± 0.48	0.55
Smoking, %	19 (25.33%)	2 (9.52%)	0.14
Heredo-colateral history of T2DM, %	23 (30.66%)	7 (33.33%)	0.79
Chronic hypertension, %	3 (4%)	0	>0.99
Gestational age, weeks, Mean, (SD)	25.95 ± 1.33	25.86 ± 1.31	0.73
Pre-pregnancy BMI, kg/m^2^, Mean, (SD)	28.54 ± 5.63	25.04 ± 3.03	0.0004
BMI, kg/m^2^, Mean, (SD)	31.14 ± 5.41	27.84 ± 3.22	0.0009
SBP, mmHg, Mean, (SD)	109.5 ± 11.89	113.8 ± 13.14	0.18
DBP, mmHg, Mean, (SD)	70.28 ± 8.39	73.86 ± 7.15	0.059
CRP, mg/dl, Mean, (min-max)	0.8 ± 0.63	0.43 ± 0.19	0.01
Fasting glucose level, mg/dl, Mean, (SD)	97.15 ± 13.5	98.33 ± 12.17	0.48
1 h glucose level, mg/dl, Mean, (SD)	176.7 ± 34.01	181.0 ± 36.93	0.76
2 h glucose level, mg/dl, Mean, (SD)	136.9 ± 34.03	149.3 ± 38.08	0.40
HbA1c, %, Mean, (SD)	5.4 ± 0.56	5.33 ± 0.35	0.89
IR HOMA, Mean, (min-max)	5.83 ± 12.23	3.85 ± 3.15	0.28
T-cholesterol, mg/dl, mean, SD	235.8 ± 45.64	246.1 ± 29.25	0.21
HDL cholesterol, mg/dl, mean, SD	65.89 ± 13.99	69.33 ± 20.71	0.89
LDL cholesterol, mg/dl, mean, SD	141.2 ± 41.98	140.7 ± 27.46	0.95
Triglycerides, mg/dl, mean, SD	226 ± 74.75	245.4 ± 75.74	0.19
TSH mU/l, mean, SD	1.72 ± 0.61	3.82 ± 0.69	<0.0001
FT4, ng/dl, SD	0.91 ± 0.2	0.96 ± 0.11	0.27

Note: SCH = subclinical hypothyroidism; T2DM = type 2 diabetes mellitus; GDM = gestational diabetes mellitus, BMI = body mass index; SBP = systolic blood pressure, DBP = diastolic blood pressure, CRP = C reactive protein; HbA1c = glycosylated hemoglobin; IR HOMA = homeostasis model of assessment for insulin resistance; T-cholesterol = total cholesterol; HDL cholesterol = high-density lipoprotein cholesterol; LDL cholesterol = low-density lipoprotein cholesterol; TSH = thyroid stimulating hormone; FT4 = free T4; SD = standard deviation.

**Table 2 biomedicines-12-02587-t002:** Maternal and newborn demographic and anthropometric characteristics, and results at birth.

Parameters	Euthyroid Group n = 75	SCH Group n = 21	*p*-Value
Gestational age, weeks, Mean, (SD)	38.35 ± 1.38	38.62 ± 1.35	0.42
BMI, kg/m^2^, Mean, (SD)	33.08 ± 5.54	30.04 ± 3.73	0.005
Cesarean section, n, %	41(54.66%)	14(66.66%)	0.45
Newborn weight, grams, Mean, (SD)	3482 ± 517.8	3408 ± 557	0.58
Ponderal index	2.27 ± 0.21	2.23 ± 0.21	0.43

Note: SCH = subclinical hypothyroidism; BMI = body mass index; SD = standard deviation.

**Table 3 biomedicines-12-02587-t003:** Maternal and perinatal outcomes.

Parameters	Euthyroid Group n = 75	SCH Group n = 21	*p*-Value
GH, n, %	13(17.33%)	6(28.57%)	0.35
Preeclampsia, n, %	0	1(4.76%)	0.21
Placental abruption, n, %	0	0	N/A
Premature delivery < 37 weeks, n, %	1(1.33%)	3(14.28%)	0.03
PPROM, n, %	2(2.66%)	0	>0.99
Macrosomia, n, %	15(20%)	2(9.52%)	0.34
Low birth weight, n, %	4(5.33%)	2(9.52%)	0.60
Fetal distress, n, %	5(6.66%)	1(4.76%)	>0.99

Note: SCH = subclinical hypothyroidism; GH = gestational hypertension; PPROM = preterm prelabor rupture of membranes.

**Table 4 biomedicines-12-02587-t004:** Multivariate regression results for premature delivery.

Parameter	OR (95%CI)	*p*-Value
Mother’s age	1.46 (0.97–2.20)	0.06
TSH	7.46 (1.04–53.36)	0.04
Pre-pregnancy BMI	1.01 (0.67–1.52)	0.93
CRP value	0.01 (0.0–131.40)	0.37
C-section delivery	0.19 (0.01–2.62)	0.21
GH	7.0 (0.49–98.6)	0.14

Note: BMI = body mass index, CRP = C reactive protein, C = cesarean, GH = gestational. Hypertension, TSH = thyroid stimulating hormone.

## Data Availability

The data that support the findings of this study are available from the corresponding author V.S. upon reasonable request.

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
