# Peer review of "Subclinical Hypothyroidism and Gestational Hypertensive Disorders in a Cohort of Romanian Pregnant Women with Gestational Diabetes Mellitus: A Pilot Study"

_biomedicines, 2024, doi:10.3390/biomedicines12112587_

Round 1
Reviewer 1 Report (Previous Reviewer 4)
Comments and Suggestions for Authors
Authors have adequately addressed concerns from the previous review and publication is recommended.
Author Response
Thank you very much for taking the time to review our manuscript, “Subclinical hypothyroidism and gestational hypertensive disorders in a cohort of Romanian pregnant women with gestational diabetes mellitus: a pilot study, “ and for your valuable suggestions.
Reviewer 2 Report (Previous Reviewer 3)
Comments and Suggestions for Authors
In response to my previous review observations, the authors have agreed to modify the few text as I pointed out. This article presents sound data with sufficient analytical support; therefore, it can be published without further review.
Author Response
Thank you very much for taking the time to review our manuscript, “Subclinical hypothyroidism and gestational hypertensive disorders in a cohort of Romanian pregnant women with gestational diabetes mellitus: a pilot study, “ and for your valuable suggestions.
Reviewer 3 Report (Previous Reviewer 2)
Comments and Suggestions for Authors
A Pilot Cohort Study among pregnant women in Romania. The author investigates whether subclinical hypothyroidism in pregnancy complicated by gestational diabetes mellitus is linked to an increased risk of gestational hypertensive disorders. The study is good, but I suggest a few comments below.
The author has made most of the suggested comments, but a few are mentioned below.
1. In the introduction section, I suggest writing about the incidence/prevalence rate of GDM and SCH in pregnant women in Europe, especially Romania.
The author replies, “Thank you for pointing this out. I found only an article from India (Dash et al.); there is no information about the prevalence of GDM and SCH in Romania in the literature (Pubmed).”
Please use those citations in the introduction section (text and reference)
Negro, R., Attanasio, R., Papini, E., Guglielmi, R., Grimaldi, F., Toscano, V., Niculescu, D. A., Paun, D. L., & Poiana, C. (2018). A 2018 Italian and Romanian Survey on Subclinical Hypothyroidism in Pregnancy. European thyroid journal, 7(6), 294–301. https://doi.org/10.1159/000490944
Chen Ailing , Luo Zouqing , Zhang Jinqiu , Cao Xiaohui, Emerging research themes in maternal hypothyroidism: a bibliometric exploration, Frontiers in Immunology, 15, 2024, DOI=10.3389/fimmu.2024.1370707. URL=https://www.frontiersin.org/journals/immunology/articles/10.3389/fimmu.2024.1370707
2. The author has defined all abbreviations in the abstract, but they should also be defined the first time and mentioned within the main text.
3. Figure 1 needs a legend.
4. Discussion was updated according to the results.
Author Response
Response to Reviewer 3
Thank you very much for taking the time to review our manuscript, “Subclinical hypothyroidism and gestational hypertensive disorders in a cohort of Romanian pregnant women with gestational diabetes mellitus: a pilot study“
To check your suggestions quickly, I will respond one by one in the following lines. In the manuscript, I underline the changes in red.
I will write your recommendation in italics and my reply in regular characters.
- 1. In the introduction section, I suggest writing about the incidence/prevalence rate of GDM and SCH in pregnant women in Europe, especially Romania.
The author replies, “Thank you for pointing this out. I found only an article from India (Dash et al.); there is no information about the prevalence of GDM and SCH in Romania in the literature (Pubmed).”
Please use those citations in the introduction section (text and reference)
Negro, R., Attanasio, R., Papini, E., Guglielmi, R., Grimaldi, F., Toscano, V., Niculescu, D. A., Paun, D. L., & Poiana, C. (2018). A 2018 Italian and Romanian Survey on Subclinical Hypothyroidism in Pregnancy. European thyroid journal, 7(6), 294–301. https://doi.org/10.1159/000490944
Chen Ailing , Luo Zouqing , Zhang Jinqiu , Cao Xiaohui, Emerging research themes in maternal hypothyroidism: a bibliometric exploration, Frontiers in Immunology, 15, 2024, DOI=10.3389/fimmu.2024.1370707. URL=https://www.frontiersin.org/journals/immunology/articles/10.3389/fimmu.2024.1370707
Thank you for pointing this out. Please read the red lines in the Introduction section.
The article:
Negro, R., Attanasio, R., Papini, E., Guglielmi, R., Grimaldi, F., Toscano, V., Niculescu, D. A., Paun, D. L., & Poiana, C. (2018). A 2018 Italian and Romanian Survey on Subclinical Hypothyroidism in Pregnancy. European thyroid journal, 7(6), 294–301. https://doi.org/10.1159/000490944
This study concerns Italian and Romanian endocrinologists’ knowledge of the new ATA guidelines on subclinical hypothyroidism diagnosis criteria and treatment guidelines in pregnancy from 2018, not the prevalence of SCH in Romanian pregnant women. According to these authors, the incidence of SCH in Europe is 10%.
- The author has defined all abbreviations in the abstract, but they should also be defined the first time and mentioned within the main text.
Thank you for pointing this out. Please read the red words in the Introduction section.
- Figure 1 needs a legend.
Thank you for pointing this out. Please read the red words below Figure 1
Reviewer 4 Report (Previous Reviewer 1)
Comments and Suggestions for Authors
This is a pilot study examining the relationship between subclinical hypothyroidism (SCH) treated with levothyroxine and gestational hypertensive disorders in Romanian pregnant women with gestational diabetes mellitus (GDM). The topic is clinically relevant and addresses an important research gap.
Major Concerns:
- Methodological Issues:
- Lack of TPOAb and iodine status assessment, which are important confounders
- No clear description of randomization process
- No blinding mentioned in the methodology
- Data Analysis:
- Multiple statistical tests performed without correction for multiple comparisons
- Limited multivariate analysis for potential confounders
Specific Recommendations:
Introduction:
- Add more recent references (several are >5 years old)
- Better justify the chosen TSH cutoff value
- Clarify the novelty of this study in the Romanian context
Methods:
- Add power calculation
- Describe randomization process
- Include TPOAb testing in future studies
- Clarify timing of levothyroxine initiation
- Add more details about follow-up protocol
Results:
- Include a CONSORT flow diagram
- Add confidence intervals for main outcomes
- Consider adjustment for multiple comparisons
- Present more detailed multivariate analyses
Discussion:
- Address study limitations more comprehensively
- Discuss potential mechanisms for increased preterm delivery in SCH group
- Compare findings more extensively with existing literature
- Strengthen clinical implications section
Tables/Figures:
- Add a CONSORT flow diagram
- Consider adding a forest plot for main outcomes
- Include supplementary tables for detailed statistical analyses
Minor Issues:
- Several typographical errors need correction
- Inconsistent formatting in references
- Some acronyms not defined at first use
Recommendation: Major revision required
Author Response
Response to Reviewer 4
Thank you very much for taking the time to review our manuscript, “Subclinical hypothyroidism and gestational hypertensive disorders in a cohort of Romanian pregnant women with gestational diabetes mellitus: a pilot study“
To check your suggestions quickly, I will respond one by one in the following lines. In the manuscript, I underline the changes in red.
I will write your recommendation in italics and my reply in regular characters.
Introduction:
- Add more recent references (several are >5 years old)
- Better justify the chosen TSH cutoff value
- Clarify the novelty of this study in the Romanian context
Thank you for pointing this out; please see the red lines from the Introduction section.
Methods:
- Add power calculation
- Describe randomization process
- Include TPOAb testing in future studies
- Clarify timing of levothyroxine initiation
- Add more details about follow-up protocol
Thank you for pointing this out; please see the red lines from Material and Methods. I have included a flow chart of study participants here.
Results:
- Include a CONSORT flow diagram
- Add confidence intervals for main outcomes
- Consider adjustment for multiple comparisons
- Present more detailed multivariate analyses
Thank you for pointing this out; please see the red lines from the Results section.
I put here a forest plot for premature delivery.
Discussion:
- Address study limitations more comprehensively
- Discuss potential mechanisms for increased preterm delivery in SCH group
- Compare findings more extensively with existing literature
- Strengthen clinical implications section
Thank you for pointing this out; please see the red lines from the Discussion section.
Round 2
Reviewer 4 Report (Previous Reviewer 1)
Comments and Suggestions for Authors
The authors have satisfactorily addressed the major concerns from the initial review. The manuscript is substantially improved in terms of methodology, statistical analysis, and discussion.
This manuscript is a resubmission of an earlier submission. The following is a list of the peer review reports and author responses from that submission.
Round 1
Reviewer 1 Report
Comments and Suggestions for Authors
This manuscript presents a pilot study examining the relationship between subclinical hypothyroidism (SCH) and gestational hypertensive disorders in pregnant women with gestational diabetes mellitus (GDM). The topic is relevant and addresses an important question in obstetric care. However, there are some limitations and areas that require improvement.Areas Requiring Revision:
- Sample size: The small sample size, particularly in the SCH group (n=13), limits the statistical power and generalizability of the findings. This limitation should be more prominently addressed in the discussion.
- Inconsistency in reporting p-values: Some p-values are reported as exact values, while others are reported as "<0.05" or ">0.05". Consistency in reporting is needed.
- Clarification on levothyroxine treatment: More details should be provided on how levothyroxine dosage was adjusted and monitored throughout pregnancy.
- Discussion of potential confounders: The authors should address potential confounding factors that could influence the results, such as iodine status or other autoimmune conditions.
- Interpretation of results: The conclusion that "subclinical hypothyroidism treated with levothyroxine in women with GDM doesn't increase the risk for gestational hypertensive disorders" should be tempered given the small sample size and limitations of the study.
- Typos and grammatical errors: There are several minor typos and grammatical errors throughout the manuscript that should be corrected.
- Tables: Consider adding a table summarizing the main outcomes (e.g., rates of gestational hypertension, preeclampsia, etc.) for both groups to improve clarity.
- Future directions: The discussion would benefit from a more detailed exploration of future research directions based on these preliminary findings.
While the small sample size limits the strength of the conclusions, the study provides a foundation for future research in this area. With minor revisions addressing the points above, particularly emphasizing the limitations and need for larger studies, this manuscript would be suitable for publication.
Comments on the Quality of English Language While the overall quality of English is good, the manuscript would benefit from a thorough proofreading to address minor errors and improve clarity in some sections. This would enhance the overall readability and professionalism of the paper.
Author Response
Response to Reviewer 1
Thank you very much for taking the time to review our manuscript, “Subclinical hypothyroidism and gestational hypertensive disorders in a cohort of Romanian pregnant women with gestational diabetes mellitus: a pilot study“
To check your suggestions quickly, I will respond one by one in the following lines. In the manuscript, I underline the changes in red.
I will write your recommendation in italics and my reply in regular characters.
1.Sample size: The small sample size, particularly in the SCH group (n=13), limits the statistical power and generalizability of the findings. This limitation should be more prominently addressed in the discussion.
Thank you for pointing this out. Please read the revised manuscript's lines 380, 381, 407, and 408.
2.Inconsistency in reporting p-values: Some p-values are reported as exact values, while others are reported as "<0.05" or ">0.05". Consistency in reporting is needed.
Thank you for bringing this to my attention. I have removed p-values ">0.05" from lines 241 and 256. This is because all the p-values above 0.05 can be found in the tables, and including them in the text would be repetitive and potentially annoying for the reader. I have chosen to keep the p-values below 0.05, as these are more important for the readers.
- Clarification on levothyroxine treatment: More details should be provided on how levothyroxine dosage was adjusted and monitored throughout pregnancy.
Thank you for this suggestion; please read lines 141-145 in the revised manuscript.
- Discussion of potential confounders: The authors should address potential confounding factors that could influence the results, such as iodine status or other autoimmune conditions.
Thank you for pointing this out; please read lines 392-397 in the revised manuscript.
5.Interpretation of results: The conclusion that "subclinical hypothyroidism treated with levothyroxine in women with GDM doesn't increase the risk for gestational hypertensive disorders" should be tempered given the small sample size and limitations of the study.
Thank you for pointing this out; please read the conclusion in the revised manuscript:
In conclusion, our results suggest that subclinical hypothyroidism treated with levothyroxine in women with GDM doesn’t increase the risk for gestational hypertensive disorders
6.Tables: Consider adding a table summarizing the main outcomes (e.g., rates of gestational hypertension, preeclampsia, etc.) for both groups to improve clarity.
Thank you for pointing this out. Table 3 summarizes the main outcomes for both groups (euthyroid and SCH), line 259.
- Future directions: The discussion would benefit from a more detailed exploration of future research directions based on these preliminary findings.
Thank you for pointing this out; please see lines 414-417.

Reviewer 2 Report
Comments and Suggestions for Authors
A Pilot Cohort Study among pregnant women in Romania. The author investigates whether subclinical hypothyroidism in pregnancy complicated by gestational diabetes mellitus is linked to an increased risk of gestational hypertensive disorders. The study is good, but I suggest a few comments below.
In the introduction section: I suggest writing about the incidence/prevalence rate of GDM and SCH in pregnant women in Europe, especially Romania.
Page 2, Line 74: “SCH in pregnancy is defined by a serum” ……… Define “SCH” for the first time before the abbreviation.
Page 3, Line 104: “in Târgu Mureș” ……………… Write the country name also.
Page 3, Line 144: “PPROM,” Define abbreviation for the first time.
As the author has referred to, the small sample size is one of the limitations.
Good Discussion
References are updated
Author Response
Response to Reviewer 2
Thank you very much for taking the time to review our manuscript, “Subclinical hypothyroidism and gestational hypertensive disorders in a cohort of Romanian pregnant women with gestational diabetes mellitus: a pilot study“
To check your suggestions quickly, I will respond one by one in the following lines. In the manuscript, I underline the changes in red.
I will write your recommendation in italics and my reply in regular characters.
I suggest writing about the incidence/prevalence rate of GDM and SCH in pregnant women in Europe, especially Romania.
Thank you for pointing this out. I found only an article from India (Dash et al.); there is no information about the prevalence of GDM and SCH in Romania in the literature (Pubmed).
Page 2, Line 51: “SCH in pregnancy is defined by a serum” ……… Define “SCH” for the first time before the abbreviation.
Thank you for pointing this out. Please read lines 51-52 in the revised manuscript.
Page 3, Line 104: “in Târgu Mureș” ……………… Write the country name also.
Thank you for pointing this out. Please read line 104 in the revised manuscript (I write the name of the country).
Page 3, Line 141: “PPROM,” Define abbreviation for the first time.
Thank you for pointing this out. Please read lines 28, 29 in the revised manuscript.

Reviewer 3 Report
Comments and Suggestions for Authors
Thank you for the opportunity to provide feedback on the research article titled "Subclinical Hypothyroidism and Gestational Hypertensive Disorders in a Cohort of Romanian Pregnant Women with Gestational Diabetes Mellitus: A Pilot Study." The study highlights the impact of SCH on pregnancy outcomes in Romanian women suffering from GHD and GDM. To improve clarity and readability, the authors should consider seeking assistance from an English editor. Based on the current observations, I recommend a minor revision.
Title: The title is appropriate.
Abstract: The layout is satisfactory.
Introduction: The introduction should be revised to reflect the context of Romania. It should be organized to address the role of the thyroid in pregnancy, the impact of GHD and GDM on pregnancy, and the link between SCH and the occurrence of GHD and GDM in pregnant women. The research gap should be clearly identified and included.
Materials and Methods:
- The section should begin with an ethical statement.
- Subheadings can be arranged as follows: Description of Study Area, Duration of Study, Inclusion and Exclusion Criteria, Diagnosis of GDM, GHD, and SCH, Anthropometric Measurements, Biochemical Analysis, etc.
- The incidence of GDM and GHD in the study population is unclear. While the study design compares normal and SCH pregnant women, the link between GDM, GHD, and SCH is missing.
Results:
- This section is generally acceptable but requires editing for better presentation. The initial lines (206-209) can be omitted.
- The results show the effects of SCH on different parameters, but they do not explain how pregnant women with GDM or GHD are affected. The data should be re-analyzed using an appropriate model.
Discussion:
- The discussion is too lengthy and repeats content from the results section.
Conclusion:
- Revise this section. The authors cannot conclude that levothyroxine does not interfere with pregnancy, as the study only observed the link between SCH, GDM, and GHD, not the effects of the medication.
Author Response
Response to Reviewer 3
Thank you very much for taking the time to review our manuscript, “Subclinical hypothyroidism and gestational hypertensive disorders in a cohort of Romanian pregnant women with gestational diabetes mellitus: a pilot study“
To check your suggestions quickly, I will respond one by one in the following lines. In the manuscript, I underline the changes in red.
I will write your recommendation in italics and my reply in regular characters.
Introduction: The introduction should be revised to reflect the context of Romania. It should be organized to address the role of the thyroid in pregnancy, the impact of GHD and GDM on pregnancy, and the link between SCH and the occurrence of GHD and GDM in pregnant women. The research gap should be clearly identified and included.
Thank you for pointing this out; please read the introduction in the revised manuscript.
Materials and Methods:
- The section should begin with an ethical statement.
- Subheadings can be arranged as follows: Description of Study Area, Duration of Study, Inclusion and Exclusion Criteria, Diagnosis of GDM, GHD, and SCH, Anthropometric Measurements, Biochemical Analysis, etc.
- The incidence of GDM and GHD in the study population is unclear. While the study design compares normal and SCH pregnant women, the link between GDM, GHD, and SCH is missing.
Thank you for pointing this out.
Please read the section Material and Method in the revised manuscript. I followed your advice.
The incidence of GDM and GHD in our region is unknown because there isn’t a universal screening for GDM and GHD in the pregnant population.
In our study, the incidence of GH in the euthyroid group was 17.33%, and in the SCH group was 30.76%.(line 252-253)
I compared pregnant women with GDM with or without SCH. I didn't compare normal and SCH pregnant women.
Results:
- This section is generally acceptable but requires editing for better presentation. The initial lines (206-209) can be omitted.
- The results show the effects of SCH on different parameters, but they do not explain how pregnant women with GDM or GHD are affected. The data should be re-analyzed using an appropriate model.
Thank you for pointing this out. I didn’t remove lines 225-228 because there, I presented the incidence of GDM and the incidence of GDM and SCH in our clinic (region) .
Discussion:
- The discussion is too lengthy and repeats content from the results section.
Thank you for pointing this out; this section is lengthy because I try to explain my results.
Conclusion:
- Revise this section. The authors cannot conclude that levothyroxine does not interfere with pregnancy, as the study only observed the link between SCH, GDM, and GHD, not the effects of the medication.
Thank you for pointing this out. Please read the conclusion in the revised manuscript. All patients with SCH were treated with levothyroxine, and TSH levels were checked every four weeks. The levothyroxine dosage was titrated to achieve a TSH level below 2.5 mU/l. What I want to say is that optimally treated SCH in patients with GDM doesn’t increase the incidence/risk for GHD. The effect of medication is normalizing the TSH levels.

Reviewer 4 Report
Comments and Suggestions for Authors
Summary
Authors investigated the effects of SCH with levothyroxine treatment in women with GDM on maternal and neonatal outcomes. The research is valuable to the field and the manuscript is well written. There are some limitations surrounding the number of participants and the interpretation of statistical outputs that are described below. For example, authors frequently describe differences between groups where it is acknowledged that there is no statistical difference. This is inappropriate. Instead, it is likely that the small sample size (n = 13 SCH + levothyroxine), which authors acknowledge in the conclusion, is not generating enough power for statistical significance which is an experimental flaw. Further, the manuscript reads throughout as if investigators are seeking to determine the effects of SCH in pregnant women with GDM. But in reality, the research is investigating the effects of women undergoing treatment for SCH (levothyroxine) with GDM. It is not truly outlining the effects of SCH in women with GDM but instead the impacts of SCH treatment. Authors are encouraged to revise the emphasis on the objectives of the study. In summary, the manuscript has the potential to have a positive impact, but with the inappropriate discussion of differences where none exist and the emphasis on SCH while disregarding the treatment, the manuscript requires major revision before publication.
Specific Comments:
Line 28 – PPROM abbreviation needs to be defined.
Line 29 – A symbol is missing for the p-value.
Line 60-61 – This sentence seems out of place.
Line 119 – 13 women with SCH seems like a low sample size given the variability in lifestyles and environments. Was a power analysis performed to ensure this number of enrollment was acceptable?
Authors define WG as weeks of gestation but are not consistent in its use.
Lines 211, 233, and 240 - In Line 233, Authors refer to a p-value of 0.057 as significant regarding BMI at birth, but in lines 211 and 240, P-values of 0.058 and 0.055 are not considered significant. This is concerning and suggests that authors are not consistent in their interpretation of significance when evaluating their statistical outputs. Authors are encouraged to establish the threshold for significance and only discuss differences where significance exists. Alternatively, authors may adopt a trend towards significance threshold at P=0.10 to discuss tendencies or trends in the data but the current inconsistencies are not appropriate.
Line 239-240 – If not statistically significant, authors are strongly discouraged from stating that premature births are higher or lower in one group compared to the other.
Line 250 – Authors acknowledge that ethnicity contributes to variation in thyroid hormone levels. Was this considered in this study? There is no mention of it during enrollment or statistical analysis.
Line 260 – Authors state that this is the first study to evaluate the impact of SCH and levothyroxine treatment on maternal and neonatal outcomes. However, the results of the 4-week interval TSH measurements are not reported; therefore, it is unclear whether the women responded similarly over time to the levothyroxine treatment as it relates to neonatal outcomes.
Line 268 – It is inappropriate to state that GDM pregnant women with SCH are older if not statistically different.
Line 289-290 – But the groups are not statistically different in age.
Line 316 – Again, authors are referring to differences where there is no statistical difference.
Author Response
Response to Reviewer 4
Thank you very much for taking the time to review our manuscript, “Subclinical hypothyroidism and gestational hypertensive disorders in a cohort of Romanian pregnant women with gestational diabetes mellitus: a pilot study. “
To check your suggestions quickly, I will respond one by one in the following lines. In the manuscript, I underline the changes in red.
I will write your recommendation in italics and my reply in regular characters.
I revised the objectives of the study as follows:
This study aimed to explore the impact of SCH treated with levothyroxine on gestational hypertensive disorders (GHD) (gestational hypertension and preeclampsia) in pregnant women diagnosed with GDM.
Line 28 – PPROM abbreviation needs to be defined.
Thank you for pointing this out; please read line 28.
Line 29 – A symbol is missing for the p-value.
Thank you for pointing this out, I write the missing symbol (>).
Line 60-61 – This sentence seems out of place.
I agree with you, thank you for pointing this out; I removed this sentence.
Line 119 – 13 women with SCH seems like a low sample size given the variability in lifestyles and environments. Was a power analysis performed to ensure this number of enrollment was acceptable?
Authors define WG as weeks of gestation but are not consistent in its use.
I agree with you. I didn’t perform a power analysis. Our study group is limited, which is why I entitled the article - a pilot study.
I used WG in the revised manuscript everywhere I used weeks of gestation.
Lines 211, 233, and 240 - In Line 233, Authors refer to a p-value of 0.057 as significant regarding BMI at birth, but in lines 211 and 240, P-values of 0.058 and 0.055 are not considered significant. This is concerning and suggests that authors are not consistent in their interpretation of significance when evaluating their statistical outputs. Authors are encouraged to establish the threshold for significance and only discuss differences where significance exists. Alternatively, authors may adopt a trend towards significance threshold at P=0.10 to discuss tendencies or trends in the data but the current inconsistencies are not appropriate.
Thank you for pointing this out. I used p=0.05 and considered all values above p=0.05 insignificant.
Line 239-240 – If not statistically significant, authors are strongly discouraged from stating that premature births are higher or lower in one group compared to the other.
Thank you for pointing this out. I rewrited the paragraph:
We found no differences between the groups regarding the incidence of GHD, premature delivery, PPROM, macrosomia, low birth weight, and fetal distress (lines 254-258).
Line 250 – Authors acknowledge that ethnicity contributes to variation in thyroid hormone levels. Was this considered in this study? There is no mention of it during enrollment or statistical analysis.
Thank you for pointing this out. Please read line 110, I write romanian ethnicity in the inclusion criteria.
Line 260 – Authors state that this is the first study to evaluate the impact of SCH and levothyroxine treatment on maternal and neonatal outcomes. However, the results of the 4-week interval TSH measurements are not reported; therefore, it is unclear whether the women responded similarly over time to the levothyroxine treatment as it relates to neonatal outcomes.
I agree with you; please read lines 140-144 in the revised manuscript:
All patients undergoing levothyroxine treatment had their TSH levels checked at 4-week intervals, and their levothyroxine dosage was adjusted according to the endocrinologist's advice.
Line 268 – It is inappropriate to state that GDM pregnant women with SCH are older if not statistically different.
I agree with you; please read lines 283,284 in the revised manuscript.
In our cohort, there are no differences between groups regarding age at the inclusion in the study.
Line 289-290 – But the groups are not statistically different in age.
I agree with you; I removed line 289-290 from the text.
Line 316 – Again, authors are referring to differences where there is no statistical difference.
I agree with you;, I removed that sentence and rewrote the paragraph as follows:
Our patient groups had no differences concerning gestational age, maternal BMI at birth, newborn weight, or ponderal index at birth. (lines 327-328)
